# The Role of RKIP in the Regulation of EMT in the Tumor Microenvironment

**DOI:** 10.3390/cancers14194596

**Published:** 2022-09-22

**Authors:** Hannah Cessna, Stavroula Baritaki, Apostolos Zaravinos, Benjamin Bonavida

**Affiliations:** 1Department of Microbiology, Immunology & Molecular Genetics, David Geffen School of Medicine, Jonsson Comprehensive Cancer Center, University of California at Los Angeles, Los Angeles, CA 90095, USA; 2Laboratory of Experimental Oncology, Division of Surgery, School of Medicine, University of Crete, 71003 Heraklion, Greece; 3Department of Life Sciences, School of Sciences, European University Cyprus, Nicosia 2404, Cyprus; 4Basic and Translational Cancer Research Center (BTCRC), Cancer Genetics, Genomics and Systems Biology Laboratory, Nicosia 1516, Cyprus

**Keywords:** RKIP, EMT, E-cadherin, N-cadherin, Snail, vimentin, laminin

## Abstract

**Simple Summary:**

Raf kinase inhibitor protein (RKIP) expression in cancer cells is significantly reduced and promoting cancer cells growth and invasiveness. Overexpresssion of RKIP has been reported to mediate pleiotropic anti-cancer activities including the inhibition of survival signaling pathways, sensitization to cell death by cytotoxic drugs, inhibition of invasion, EMT and metastasis. The molecular mechanism by which RKIP inhibits EMT is not clear. In this review, we have examined how RKIP inhibits the selected EMT gene products (Snail, vimentin, N-cadherin, laminin alpha) and found that it involves signaling cross-talks between RKIP and each of the EMT gene products. These findings were validated by bioinformatic analyses demonstrating in various human cancers a negative correlation between the expression of RKIP and the expression of the EMT gene products. These findings suggest that targeting RKIP induction in cancer cells will result in multiple hits by inhibiting tumor growth, metastasis and reversal of chemo-immuno resistance.

**Abstract:**

The Raf Kinase Inhibitor Protein (RKIP) is a unique gene product that directly inhibits the Raf/Mek/Erk and NF-kB pathways in cancer cells and resulting in the inhibition of cell proliferation, viability, EMT, and metastasis. Additionally, RKIP is involved in the regulation of cancer cell resistance to both chemotherapy and immunotherapy. The low expression of RKIP expression in many cancer types is responsible, in part, for the pathogenesis of cancer and its multiple properties. The inhibition of EMT and metastasis by RKIP led to its classification as a tumor suppressor. However, the mechanism by which RKIP mediates its inhibitory effects on EMT and metastases was not clear. We have proposed that one mechanism involves the negative regulation by RKIP of the expression of various gene products that mediate the mesenchymal phenotype as well as the positive regulation of gene products that mediate the epithelial phenotype via signaling cross talks between RKIP and each gene product. We examined several EMT mesenchymal gene products such as Snail, vimentin, N-cadherin, laminin and EPCAM and epithelial gene products such as E-cadherin and laminin. We have found that indeed these negative and positive correlations were detected in the signaling cross-talks. In addition, we have also examined bioinformatic data sets on different human cancers and the findings corroborated, in large part, the findings observed in the signaling cross-talks with few exceptions in some cancer types. The overall findings support the underlying mechanism by which the tumor suppressor RKIP regulates the expression of gene products involved in EMT and metastasis. Hence, the development of agent that can selectively induce RKIP expression in cancers with low expressions should result in the activation of the pleiotropic anti-cancer activities of RKIP and resulting in multiple effects including inhibition of tumor cell proliferation, EMT, metastasis and sensitization of resistant tumor cells to respond to both chemotherapeutics and immunotherapeutics.

## 1. Introduction

The epithelial-mesenchymal transition (EMT) is a distinctive event occurring in many types of normal and cancerous cells where epithelial cells are transiently transformed into motile mesenchymal cells [1,2,3]. The mesenchymal cells can also return to their epithelial state through an opposite process, the mesenchymal-epithelial transition (MET) [1,2]. EMT was firstly characterized as an essential process for embryonic development, where epithelial cells differentiate and migrate to their final destinations as mesenchymal cells, thus disrupting the apical-basal polarity of typical epithelial cells [2,4,5]. Mesenchymal cells are known for their independent and motile nature and invasive capabilities, as well as for lacking binding through adhesive molecules, polarity, and direct cell–cell contact [3,6,7].

In general, there are 3 types of EMT: In type 1 EMT occurs during development, in type 2 EMT facilitates wound healing, while type 3 is so called oncogenic EMT and refers to cancer cells [8]. There has also been evidence that EMT is not a strict transformation, but a transition with intermediates with both epithelial and mesenchymal phenotypes [9]. EMT results in a more cell motility as the cell transitions from an epithelial cell with tight attachments to the basement membrane to a mesenchymal phenotype through the activation of transcription factors, reorganization of the cell, and expression of certain proteins that confer motility [2,10]. While epithelial cells are generally uniformly organized into sheets with clear apical and basal sides of the sheet defined through the polarization of the individual cells, mesenchymal cells have irregular shapes and front-to-back polarity without the formation of uniform structures, distinctly defining the beginning and end formations of EMT [11,12].

While EMT types 1 and 2 are essential to wound healing and embryonic development, respectively, the oncogenic EMT is utilized in cancer progression to transform epithelial cells into mesenchymal cells with reduced cell adhesions, thus being able to migrate through the extracellular matrix (ECM) and acquire invasive characteristics [13]. Cancer cells hijack the normal EMT process to adapt an invasive and motile phenotype, that allows dissociation from the original tumor site and rise of metastases throughout the body, while avoiding apoptosis [9,14,15,16]. Once relocated, the cancer cells undergo MET, establishing a new site of tumor growth away from the original position [17]. The rise in EMT and cancer progression is due to activation of specific transcription factors, cytoskeleton reorganization, loss of cell polarity and movement through the extracellular matrix, and loss of adhesion mechanisms [6,15,18,19,20].

## 2. Epithelial vs. Mesenchymal Cells

### 2.1. Epithelial Cells

Epithelial cells serve as a physical barrier between the environment and our internal systems, creating the first line of defense and contact against microbes and invaders. Epithelial cells are known for their polarity along the apical-basal axis, apical-basal polarity, that accounts for the asymmetry in epithelial cells that is required for normal cell function [21,22]. Epithelial cell function requires the polarity and asymmetrical organelle organization for directional transport of molecules from the external environment into the body [23]. A second axis of planarity that is required for the typical epithelial cell phenotype is planar cell polarity (PCP), which is the plane of organization of cells in the epithelial tissue, an axis of organization that is also common in mesenchymal cells [21,24]. PCP proteins are shuttled by microtubules to the junction locations to create the asymmetric epithelial phenotype and to establish the PCP signaling pathway that helps in polarizing other cellular bodies, such as cilia orientation [25].

Cell polarity is further established through interactions between the cytoskeleton and adhesion molecules in order direct cell functions in a specific orientation [24]. In particular, the minus ends of microtubules are stabilized by the interaction with adherence molecule E-cadherin, creating a molecular signaling pathway across the cell and between neighboring cells [26]. With connection through adherens junctions, the PCP signaling pathway aids in establishing the planar phenotype in other cells, though the exact mechanism of this widespread event remains unclear [27]. The establishment of this signaling pathway adds to and maintains the polar phenotype of epithelial cells by continuing contact and communication between neighboring epithelial cells [26,28]. Cell contact through cadherin molecules initiates Cdc42 signaling, which aids in organizing the polarity of the cell through protein complexes, such as Par at the E-cadherin tight junctions [29,30,31]. Active Cdc42 signaling is necessary for Par6 localization to the apical membrane for its role in coordinating polarization, but it is also involved in maintaining tension levels for continued cell–cell contact, which is necessary for the tissue wide organization [32,33]. Polarized epithelial cells organize into tight monolayers with the apical side facing the outside environment and the basal side facing inward, mediating ion and substance exchange between the body and the outside compartment [34].

Epithelial cells, with their distinct polarity, have different cell structures dependent on their localization in each side of the cell. The apical domain is distinguished by the microvilli while the basal side is important for its connections to the interior environment through the interaction between integrin receptors and the extracellular matrix, providing signaling and transport across the cell [22,35]. Extending filopodia from epithelial cells also creates a network of communication and a tight layering of cells [22]. The filopodia connect to the extracellular matrix, acting as a guide for epithelial migration involved in wound healing and maintenance of the epithelial lining [36,37].

Connection to the extracellular matrix and cellular-communication is necessary for cell survival in epithelial cell lines and confers resistance to apoptosis [38,39]. Adherens junctions, composed of E-cadherins in epithelial cells, serve as an adhesion point between epithelial cells, and the adhesive and interactive nature of these molecules are a response to the extracellular calcium levels [40,41]. E-cadherin organizes into a zipper formation that creates the typical tight seal of the epithelium, and since E-cadherin does not bind to other types of cadherins such as N-cadherin, the type of molecule at the cell–cell junction is important to maintenance of the epithelial layer [40,41]. E-cadherin is linked to the cytoplasmic domain through adhesion to α-catenin (a cytoplasmic plaque domain involved with actin-binding) though the E-cadherin cytoplasmic tail, while β-catenin links α-catenin to the cytoplasmic domain, playing a role in regulation and signal transduction by completing the connection to the domain [41,42,43]. The actin filament polymerization and organization by vasodilator stimulated phosphoprotein (VASP) (a protein involved focal contacts between cells) within the cytoplasm is necessary for E-cadherin function in adhesion and intracellular communication, essential aspects of the epithelial layer [41].

### 2.2. Mesenchymal Cells

Mesenchymal cells are motile, invasive, apoptosis-resistant cells that lack a strict organization [44]. Mesenchymal cells are required for early embryonic development and organ and tissue localization [45,46]. Incorrect regulation of mesenchymal cells can lead to destructive outcomes for the body including chronic inflammation and tumor progression [6,20]. The mesenchymal phenotype lacks the apical-basal polarity and accompanying cell–cell contacts that are typical of epithelial cell types; however, the mesenchymal cells develop a front-back polarity that allows for their migratory function [47]. The front-back polarity develops through the migration of certain molecules toward the invasive front of cancer cells such has the increase in αvβ3-integrin as a result of induction by Snail1 [3]. In addition, the front-back polarity phenotype establishes directionality in migrating cells, extending actin filaments including lamellipodia and filopodia toward the front edge of the cell to establish turning mechanisms [48,49]. To establish the trailing back of the migrating mesenchymal cell, myosin IIB localized in the rear and activated by MLC phosphorylation establishes actin bundles and stable adhesions that inhibit the formation of protrusions that extend in the front of the cell [50]. The resulting phenotype is a spindle-like shape with an up-regulation of mesenchymal molecules such as N-cadherin, vimentin, and fibronectin [46].

The spindle-like morphology of the mesenchymal cells differentiates from the more polygonal, compact shape of epithelial cells [51]. During EMT, the cytoskeleton rearranges in order to provide movement abilities to the mesenchymal cell, degrading cell-ECM interactions through production of metalloproteinases (MMPs) that work to degrade the ECM [6,52]. The change into a spindle structure is partially controlled by the downregulation of syndecans, but the exact mechanics of this process is relatively unknown [51,53]. Mesenchymal cells exhibit fibroblast features after undergoing EMT as shown through a study of hepatic cells, causing fibrosis when EMT activity is left unchecked [54]. As discussed earlier, myosinIIB activation is essential in formation of a tail end of migrating mesenchymal cells, creating a clear front and back directionality in mesenchymal shape [50]. In addition, mesenchymal cells resulting from EMT acquire filopodia made of actin bundles on the front end that aid in movement and turning as well as ventral stress fibers that run between focal adhesions [49,55]. Overall, the regulation of mesenchymal shape and the cellular structures that govern the shape is extremely complex and yet not clearly elucidated at the molecular level, thus providing new challenges on mesenchymal cell research.

## 3. Overview on Oncogenic EMT: EMT-Associated Biomarkers

EMT is thought to be the major force behind cancer aggressiveness, as it promotes cancer immunoescape and formation of a cancer stem cell phenotype, thus suggesting that EMT is rather involved in tumor invasiveness and stage progression than in oncogenesis [56]. As such the molecular regulation of oncogenic EMT is mediated by several coding and non-coding transcripts that are implicated in multiple signaling pathways associated with tumor progression and aggressiveness.

Among the involved EMT regulators are several non-coding transcripts including members of the miR-200 family of microRNAs. The downregulation of miR-200 is frequently observed in many cancer types, and has been attributed to negative signaling by TGFβ, which in turn leads to reduced expression of the epithelial hallmark marker E-cadherin, thus suggesting the miR200 as an EMT suppressor [57]. PTEN also plays an inhibitory role in oncogenic EMT through its suppressive effects on AKT signaling. PTEN dephosphorylates phosphatidylinositol-3,4,5-triphosphate (PIP_3_) to phosphatidylinositol-4,5-biphosphate (PIP_2_), inhibiting AKT activation by PIP_3_ and associated stimulation of cell proliferation [58]. The PcG protein B lymphoma Mo-MLV insertion region 1 homolog (Bmi-1) downregulates PTEN function by binding to PTEN, which activates the AKT pathway, and by activating this signaling pathway, E-cadherin is downregulated, allowing the cells to be more motile and promoting the progression of the EMT phenotype [59,60]. In breast cancer preclinical models, cells lacking PTEN acquire mesenchymal characteristics, as evidenced by the high levels of the mesenchymal marker vimentin and the loss of E-cadherin at adhesion sites, as well as by the increase of the EMT-inducing transcription factors Snail1, Slug, ZEB1, and Twist2 [61]. In contrast, the restoration of PTEN reversed the mesenchymal phenotype, thus allowing the epithelial characteristics to reemerge [61]. Abi1 is a substrate of PTEN that is dephosphorylated by PTEN, deactivating Abi1. In PTEN knockout experiments however, Abi1 is overexpressed and induces the expression of EMT transcription factors and the mesenchymal phenotype, demonstrating EMT regulation by PTEN and its substrates [61].

Certain biomarkers currently used in cancer diagnosis are indicative of EMT progression in multiple cancer types [62]. The EMT-associated biomarkers come from a wide range of proteins, including cell surface proteins such as E-cadherin, N-cadherin and integrins, cytoskeletal proteins such as vimentin and β-catenin, transcription factors like Snail1, Snail2, twist and LEF-1 as well as extracellular matrix proteins such as fibronectin and laminin (CS Scanlon et al., 2013). In the following subsections we analyze the major EMT-inducing and -suppressing biomarkers with clinical significance, as well as their molecular regulation and interplay.

### 3.1. Clinically Relevant EMT-Inducing Biomarkers and Their Regulation

#### 3.1.1. Vimentin

Vimentin is part of the intermediate filament family of proteins, playing a role in cytoskeleton arrangements, organelle organization, and signal transduction [63]. Vimentin is upregulated during developmental and oncogenic EMT, resulting in a mesenchymal cell phenotype and promoting the characteristic motility of the mesenchymal cells [63]. In knock out studies in mice, loss of vimentin was shown to be associated with a loss of mechanical stability and strength as well as reduced mobility and reduced contractile ability, showing vimentin to play an essential role in movement and stability in mesenchymal cells where vimentin is upregulated [64,65]. Motility is promoted by vimentin expression through induction of increased turnover of focal adhesion points and loss of cell–cell contacts, regulating microtubule development during migration that protects against compression in cramped cellular spaces [66]. Vimentin works with microtubules to form the cytoskeleton network, and is often used as a biomarker for EMT due to its upregulation in mesenchymal cells [67]. Vimentin is a key regulator of cytoskeleton organization and therefore, it is believed to be correlated to the increased motility and invasiveness in mesenchymal cells [68]. In addition, vimentin is related to poor prognosis in many types of cancers and is shown to be a downstream effector of the slug signaling pathway that plays a role in EMT progression [68].

Vimentin regulation is multifaceted and is achieved at different levels through several different pathways, as summarized below:Transcriptional Regulation

Though the mechanism of Vimentin regulation is unclear because the protein usually acts as a transcriptional repressor, Smad interacting protein-1 (SIP1) has been shown to increase Vimentin mRNA and protein expressions, indicating that it most likely regulates vimentin expression at a transcriptional level [69]. TGFβ1 affects another SMAD protein, SMAD3, which in turn interacts with vimentin’s AP-1 elements when a SMAD binding site is missing, allowing for the recruitment of p300 [70]. One theorized role for p300 is as a co-activator of vimentin expression, binding transcription factors to the vimentin promoter and promoting expression [70]. TGFβ also induced EMT and increases the vimentin expression level through the activation of SMAD4 and its repression of ARG2, a promoter of the epithelial phenotype, and it has been suggested that the mitogen-activated protein kinase (MAPK) signaling cascade is essential for the transcription of the SMAD pathway involved [71].
Epigenetic Regulation

As discussed in the previous section, although p300 has been shown to be involved in vimentin expression through the TGFβ1 and SMAD3 signaling, the exact role of p300 has yet to be determined, leading to a possible theory for p300 to have a role in increased acetylation and opening chromatin structure for increased vimentin transcription [70]. Vimentin epigenetics are also shown through the role of methylation. The vimentin gene is frequently methylated in advanced colorectal cancer cells, allowing methylation of the gene to be used as a diagnostic tool for diagnosis of late-stage colorectal cancer [72]. In hypoxia-induced EMT, the regulation of vimentin occurs through activation of HIT-1α which activates HDAC3, deacetylating the vimentin gene and leading to an increase in the EMT phenotype [73].
Post-Transcriptional Regulation

Vimentin is regulated at the post-transcriptional level by multiple microRNAs. miR-548a and miR-22 are direct vimentin suppressors, and associated with inhibition of vimentin-mediated invasiveness and E-cadherin upregulation, respectively [66]. MiRNA146a is usually found downregulated in squamous cell carcinoma (ESCC), resulting in increased cancer cell mobility and tumor progression via upregulation of vimentin [74]. In colorectal cancer, microRNA-17-5p serves as a suppressor of EMT, through direct targeting of vimentin and inhibition of the Wnt pathway [75]. Long intergenic non-coding RNAs (lincRNAs) have been also been involved in the regulation of vimentin by affecting the expression of microRNAs with a regulatory role on vimentin expression. For example, LINC01488 has been shown to increase the expression of miR-124-3p and miR-138-5p, both of which have been linked with the direct post-transcriptional repression of vimentin [76]. On the flip side, vimentin may also regulates other microRNAs with tumor suppressing function, thus promoting increased invasiveness and other tumor-associated properties pro when it is present in mesenchymal cells [66]. MicroRNA146a, which targets vimentin, has been found to repress the tumor progression in esophageal squamous cell carcinoma (ESCC), and when MiRNA146a is repressed, ESCC mobility is increased through the expression of the mesenchymal marker vimentin, showing the regulatory effect of MiRNA146a on vimentin [74]. Shown in colorectal cancer, microRNA-17-5p also serves as a regulator of vimentin, and as an extension EMT, through targeting of the 3′UTR of the gene for vimentin VIM, repressing the expression of vimentin as well as repressing EMT through the inhibition of the Wnt pathway [75].
Post Translational Regulation

Vimentin is modified post-translationally in several ways such as citrullination in macrophages (arginine residues changed to citrulline by peptidylarginine deiminase), sumoyation, and O-GlycNAcylation (to maintain structure and a scaffold for migration) [77]. Citrullination occurs on extracellular vimentin through peptidyl arginine deiminases, converting arginine to citrulline which may result in autoimmune diseases as the immune system tags the vimentin as foreign [78]. In addition, vimentin is regulated by certain kinases through phosphorylation that affects its functional capabilities [77]. Its phosphorylation sites include sites that interact with p21-activated kinase (PAK), cAMP-dependent protein kinase A (PKA), Rho-associated protein kinase (ROCK), cyclin-dependent kinase 1 (CDK1), and several others [78]. Phosphorylation of vimentin usually inhibits its ability to function and assemble. PKA has been shown to phosphorylate vimentin at the sites S38 and S72, thus affecting the filament formation, while p-21 activated kinase (PAK) phosphorylates several sites, that affect vimentin’s structural reorganization [77]. In contrast, phosphorylation by mitogen-activated protein kinase activated protein kinase-2 (MAPKAP-KII) at Ser38, Ser50, Ser55, and Ser82, is thought that help vimentin to serve as a sink for phosphate groups [78].

#### 3.1.2. N-Cadherin

N-cadherin is attached to the cell surface through the cytoplasmic tail extending into the cytoplasm that attaches to β-catenin which attaches to α-catenin to form the cadherin-catenin adhesion complex [79]. The mesenchymal phenotype during EMT is characterized by a loss of the epithelial E-cadherin and a gain of N-cadherin in a “cadherin switch”, leading to a loss of cell–cell contact and an increase in cell adhesiveness as observed in ovarian cancer progression [79,80]. While E-cadherin is known to suppress the Wnt/β-catenin and the RTK/P13K pathway, N-cadherin activates the MAPK, ERK, and PI3K pathways in order to promote cell migration and survival [79]. In analysis of pancreatic cancer cells, the overexpression of N-cadherin promotes the EMT process through the activation of the ErbB signaling pathway, creating possible new therapeutic targets for pancreatic cancer cells [81]. Silencing of N-cadherin generates a less invasive and motile phenotype in cancer cell lines along with the down regulation of fibronectin and vimentin and increased expression of the epithelial marker E-cadherin, losing the stem cell-like features of mesenchymal cells [81]. In contrast the loss of E-cadherin, and the accompanying replacement by N-cadherin, known as cadherin switch, proves to be an influential force in establishing the mesenchymal phenotype because anchoring and communicative properties of E-cadherin to neighboring cells is lost, giving cells the ability to relocate from the original site and invade other tissues or organs such as in the progression of cancer [57,82].

In many cancer types, increased N-cadherin is shown to be correlated with tumor aggressiveness and lower survival rates, though the exact mechanisms of N-cadherin involvement in the above processes have not been clearly elucidated [83,84]. Accordingly, N-cadherin has been found to be a good target biomarker for cancer progression, due to its presence on both tumor cells and the circulating serum of cancer patients [84]. In general lines, the cadherin switch from E-cadherin to N-cadherin expression during EMT has been associated with increased tumor invasiveness and seems to be a critical step in cancer progression, resulting from a loss of stable adhesion points between epithelial cells [85].

TGF-β has been considered a major regulator of the cadherin switch, thus increasing N-cadherin levels over E-cadherin during EMT [84,86]. β-catenin plays a role in N-cadherin regulation through the downregulation of E-cadherin [87,88]. The downregulation of E-cadherin releases free β-catenin that is able to activate the TCF/LEF-1 that are involved in the initiation of EMT in epithelial cells expressing integrin-linked kinase, leading to the upregulation of N-cadherin through the process of EMT [87]. The detailed regulation of N-cadherin at different levels is mediated through multiple pathways as analyzed below:Transcriptional Regulation

N-cadherin is transcriptionally regulated through mechanisms of down-regulation of E-cadherin. The PI3K/PTEN pathway is a key regulator of this switch through the upregulation of Snail and Twist as well as a loss of PTEN activity [89]. PTEN knockdown causes a large increase in the expression of both Snail and Twist which have a role in the switch from E-cadherin to N-cadherin in melanoma cells [89]. Other regulatory factors also affect N-cadherin expression at the transcriptional level. NF-*κ*B, demonstrated in hepatocellular carcinoma, increases the transcription of N-cadherin and decreases that of E-cadherin through the NF-*κ*B signaling pathway when it is activated by receptor activator of nuclear factor kappa B ligand (RANKL) [84]. However, the exact role of NF-*κ*B in N-cadherin activation is unknown since silencing of NF-*κ*B through siRNA results in decreased cancer invasiveness, but N-cadherin is still upregulated [84]. Snail and Slug have also been connected as transcription factors that promote the expression of N-cadherin through separate signaling pathways [90].
Epigenetic Regulation

Epigenetic factors also play a key role in the progression of EMT. Twist, a transcriptional regulator of N-cadherin, also interacts with SET8, a histone-methyltransferase, and the interaction between the two recruits SET8 to the promoter region of N-cadherin, methylating H4K20 which activates the expression of N-cadherin in EMT [73]. As discussed with Vimentin, HIT-1α activates HDAC3, deacetylating H3K4 in order to activate the expression of the N-cadherin gene and promote the associated mesenchymal phenotype [73].
Post-Transcriptional Regulation

miRNAs are very involved with the post transcriptional regulation of N-cadherin. miRNA-122, which is known to play a large role in the liver as a tumor suppressor, is regulated by the Wnt/β-catenin pathway, and in hepatic stellate cells, miRNA-122 is shown to have a negative correlation with N-cadherin, indicating that this microRNA plays a role in repressing the expression of N-cadherin [91]. Similarly, in gastric cancers, miR-145-5p upregulation reduced the expression of N-cadherin, leading to the assumption that miR-145-5p targets N-cadherin in the regulation of EMT [92]. miRNA-145 also showed a similar repressive role in bladder cells, inhibiting migration through targeting N-cadherin and corresponding downstream molecules such as matrix metalloproteinase-9 [93]. Non-small cell lung cancer transfected with a mimic of miR-148b showed a similar repressive function of N-cadherin, but when an miR-148b inhibitor is introduced, N-cadherin expression is increased, showing the role of this microRNA in targeting and repressing N-cadherin [94]. miR-199b-5p serves as another example of N-cadherin post transcriptional regulation shown in hepatocellular carcinoma as the microRNA targets the 3′UTR of N-cadherin, resulting in decreased expression of N-cadherin and showing a negative correlation [95].
Post-Translational Regulation

Regulation of N-cadherin can also occur through the post-translational modification of another molecule that induces the production of N-cadherin. In two separate post-translational events, ubiquitin specific peptidase 51 (USP51) specifically binds ZEB1 and COP9 signalosome subunit 5 (CSN5) deubiquitinates ZEB1, stabilizing this protein in order to promote the expression of N-cadherin.

#### 3.1.3. EMT-Inducing Transcription Factor (EMT-TF) Snail

EMT-TFs have been implicated in the positive regulation of oncogenic EMT in several cancer types and are considered of high clinical significance. For example, progressive neck squamous cell carcinomas (HNSCC) with EMT phenotypes are characterized by overexpression of the EMT-TFs Twist1, Twist2, Snail1 and Snail2 [96]. In a meta-analysis, the elevated levels of these EMT-TFs in HNSCC were correlated to lower patients’ overall survival (OS), with EMT to account for tumor invasiveness, thus suggesting that the expression of the aforementioned transcription could serve as EMT biomarkers in HNSCC and other cancer types [96].

Snail, a major EMT-TF, induces EMT mainly through suppression of the E-cadherin expression [15]. Snail is a member of the zinc-finger transcriptional repressor family, which is involved in embryonic development through the formation of the mesoderm. However, in early development, Snail inducing effects on EMT is replaced by vimentin which in turn represses E-cadherin, thus decreasing the adhesion points of the epithelial cells and allowing a mesenchymal phenotype to prevail [97]. The zinc finger domains of the Snail family of proteins are extremely conserved and they bind to E box promoters to repress the gene of interest [98,99]. The complex molecular regulation of Snail is described below:Transcriptional Regulation

Snail is transcriptionally regulated through several signaling pathways. Receptor Tyrosine Kinase pathways induce Snail expression through inhibiting GSK-3β which is an inhibitor of Snail function [97,100]. The inhibition of the GSK-3β inhibitor can be established through Wnt binding to its receptor Frizzled, giving the signal to inhibit GSK-3β and increase migratory transcription factors, allowing Snail to avoid inhibition and phosphorylation by GSK-3β [100]. NF-*κ*B binds to the promoter of Snail and promotes its transcription through signaling from TNFα, leading to Snail transcription which is mediated by CSN2 which blocks the ubiquitylation and degrading by GSK-3β [15]. In a study of Hepatocarcinoma cells (HCC), cells treated with valproic acid (VPA) have been shown to increase the protein expression of Snail through the increase in mRNA expression through the activation of NF-*κ*B by VA, which interacts with the promoter for Snail to increase expression [101].
Epigenetic Regulation

Snail plays a large role in the epigenetic regulation of other genes that are essential for EMT. First, Snail has been found to recruit histone deacetylases (HDACs) to the E-cadherin promoter site, maintaining histone methylation and repressing E-cadherin gene expression [102]. In addition, Snail is involved with heterochromatin formation and repression of E-cadherin transcription through association with G9a at the E-cadherin promoter, which is responsible for H3K9 demethylation [102]. Suv39H1 also interacts with Snail in the repression of E-cadherin, forming a trimethylation of H3K9, a marker of constitutive heterochromatin and being accompanied by a decrease in acetylation of H3K9 [102]. Snail has an overall role in the expression of E-cadherin through epigenetic modifications. It is unknown if Snail itself is affected by epigenetic regulation.
Post-Transcriptional Regulation

RNA m^6^A methylation is a post-transcriptional modification that occurs at the nitrogen-6 position of adenine. m^6^A RNA methylation of Snail contributes to increased Snail translation and folds the protein at a higher rate through the binding of YTHDF1 and EIF3 to m^6^A-modified RNAs that increases the affinity for polysomes [103]. In addition, miRNAs play an important role in post-transcription regulation of Snail. Some miRNA, such as miRNA-210, promote migration and cancer progression in breast cancer stem cells through targeting Snail, while in contrast, some other, such as miR-34c, negatively regulate Snail and decrease EMT progression [104]. miR30b has been shown to have a negative correlation with Snail expression, being shown to reduce the translation of Snail when this microRNA interacts with the 3′UTR, leading to repression of migration mechanisms in pancreatic ductal adenocarcinoma [105]. miR-124 inhibits Snail signaling in the AKT/GSK-3β/Snail signaling pathway, leading to decreased EMT and migration [106]. miR-22 has been shown to directly target Snail as shown in gastric cancers, decreasing the progression of EMT, while the repression of miR-22 correlates to increased expression of Snail [107].
Post-Translational Regulation

Post-transcriptionally, the Snail protein is stabilized through BRD4 blocking the degradation properties of the proteasome, not allowing Snail to be degraded [108]. Similarly, Snail is blocked from degradation through phosphorylation by p21-activated kinase (PAK1), which induces the nuclear localization of Snail, increasing its ability to function and avoiding ubiquitin-mediated proteasome degradation [97]. Another pathway of ubiquitination-blocking was discovered through radiation treatment, where researchers saw an increase in the stabilization of Snail and a decrease in its degradation through an increase in the expression of COP9 signalosome 2 (CSN2), which blocks ubiquitination [109]. G-protein-coupled Estrogen Receptor (GPER) also plays a role in port-translational regulation, since it has been found to destabilize Snail and therefore, downregulate its expression through increased expression of FBXL5, though the exact mechanism has yet to been determined [110]. In HCC cells, VPA was also shown to stabilize Snail through phosphorylation of GSK-3β and AKT, leading to upregulation of Snail [101]. Demonstrated in CRC cells, Cten also regulates and stabilizes Snail through inhibiting protein degradation, so Snail can localize in the nucleus and act as a transcription factor for migration genes, promoting a motile phenotype that supports cancer invasiveness [111]. Overall, post-translationally, Snail is regulated through its ability to avoid ubiquitination and degradation by proteasomes in the cytoplasm of the cell.

#### 3.1.4. Epithelial Cell Adhesion Molecule (EpCAM)

EPCAM has been considered a prognostic biomarker, as its overexpression has been linked to cancer development, whereas normal levels are hallmarks of normal epithelial cell function [112]. EpCAM serves mainly as a homophilic cell adhesion molecule, but it has also been linked to roles in proliferation, differentiation, and cell-migration with differing levels of expression [113]. In the early stages of embryonic development, EpCAM is not expressed solely by epithelial cells, indicating a greater role of EpCAM plasticity and early development, and through the use of knock-out studies in Zebrafish, loss of EpCAM is linked to lethality and loss of tissue integrity, displaying the dual role of EpCAM in adhesion and motility [114,115]. EpCAM also plays a role in the assembly of adherens junctions as shown by the dramatic decrease in mRNA and protein expression of nectin1, a molecule involved in regulating the assembly of adherens and tight junctions, in EpCAM mutant mice [116]. EpCAM is a type I transmembrane superficial glycoprotein found in low levels on the basolateral surface of normal epithelial cells, but increases in EpCAM levels during cancer development make it an essential biomarker for prognosis of many cancer types affecting epithelial cells, making EpCAM levels an important marker of epithelial cell function [117].
Transcriptional Regulation

Some transcription factors have been correlated to the expression levels of EPCAM, leading to a probable role in transcriptional regulation of EPCAM through these factors. For example, in metastatic lymph nodes in several cancer types, the increased expression of Esx/Elf3 correlated to an increased expression of EPCAM [118]. In addition, in ovarian cancer, AP2α, Ets1, Ets2, E2F2, E2F4, and STAT3 transcription factors have all been shown to interact with the EPCAM gene, showing some probable level of regulation of the EPCAM gene by these factors at the transcriptional level [118]. Wild type p51 has also been shown to bind to intron 4 of EPCAM, meaning that it also plays a role in the transcriptional regulation of EPCAM [118].
Epigenetic Regulation

The epigenetic regulation of EPCAM relies significantly on the amount of methylation on the CpG islands of the promoter sequence. Low levels of methylation at the promoter sequence have been shown to be associated with high levels of EPCAM expression, while on the other hand, hypermethylation correlated to a lack of EPCAM expression [119]. However, regulation by methylation seems to differ between tissue types in cancer progression. In breast cancer, EPCAM expression did not seem to be correlated to EPCAM promoter methylation, while in colon cancer tissue, the unmethylated promoter sequence is correlated to a 1000 fold increase in EPCAM expression in comparison normal tissue [119]. Expression of EPCAM has also been shown to be associated with histone modifications, including an acetylated histone 3, acetylated histone 4, and trimethylation of lysine 4 of histone 3 [120].
Post-Transcriptional

A lot is not known about the post-transitional regulation of EPCAM through miRNA. miR-181, miRNA200c, and miRNA205 have each been shown to be involved in the upregulation of EPCAM, but the exact mechanism as how the microRNA interact, direct or indirect mechanism, with EPCAM is not known [118].
Post-Translational Regulation

EPCAM is post-translationally modified in many ways, meaning its regulation as a protein relies on each step. Proteolytic cleavage is an essential step to the processing of the EPCAM protein in order to ensure correct function [121]. The N-terminal signal peptide of EPCAM is cleaved after Ala23 by a signal peptidase and between Arg80 and Arg81 by serine and cysteine proteases [121]. In addition, EPCAM is cleaved by regulated intramembrane proteolysis (RIP) to allow it to shed the extracellular domain of EPCAM, EpEX, followed by another cleavage to release the EpICD [121]. Three glycosylation sites also exist on EPCAM at Asn74, Asn111, and Asn198 (essential for EPCAM stabilization) which have been shown to be essential to EPCAM’s role in EMT [121]. More glycosylation sites and post-translational modifications have been found to occur to EPCAM, but the function of these modifications and their role in EPCAM’s function have yet to be found [121].

## 4. Clinically Relevant EMT-Suppressing Biomarkers and Their Regulation

### 4.1. E-Cadherin

E-cadherin serves as a defining feature and biomarker of epithelial cells and is regulated by many molecules that interact with E-cadherin at the adherens junctions. E-cadherin is an adhesive molecule that provides an essential interaction between epithelial cells, typical of epithelial cell presentation, but E-cadherin is lost during the EMT transition, leading to the ability for cell movement [122,123]. The regulation of E-cadherin begins with the structural components of this molecule as its intercellular tail protrudes into the epithelial cell cytoplasm and interacts with β-catenin and α-catenin, forming a connection to the actin-myosin network [124]. E-cadherin on the cell surface connects with β-catenin, which in turn connects to α-catenin, creating a network between actin and the E-cadherins with α-catenin as a linker, establishing communication between the outside environment and cellular components [43,125]. Modification of these molecules can, in turn, affect the phenotype of the epithelial cell. Vinculin is a protein that binds actin and is involved in the adherens junctions through interactions with α-catenin [126]. α-catenin is subject to forces originating through neighboring cells, converting the mechanical force into a chemical response as it reveals its vinculin binding site and more vinculin is recruited, acquiring more actin connections to stabilize the cell against the outside forces [126].
Transcriptional

Loss of E-cadherin in mesenchymal cells occurs through EMT related transcription factors such as Snail, Slug, and Twist that suppress e-cadherin; however, the loss of e-cadherin on the surface does not necessarily contribute to invasiveness in mesenchymal cells as it has been found that the activity state of e-cadherin is required for function of epithelial cells and the loss of activity, despite high levels of e-cadherin, can contribute to a motile phenotype [127,128]. Regulation of E-cadherin during EMT seems to be influenced by multiple factors including transforming-growth factor β (TGFβ), vimentin, and nuclear factor kappa B (NF-*κ*B) [14,129]. TGFβ is a transcription factor highly involved with cell-mediated apoptosis and, in and experiments where TGFβ signaling is active, E-cadherin expression is inhibited in epithelial cells and EMT is induced [14,130]. Several pathways have been proposed using TGFβ signaling as an inducer for EMT, although the cooperativity seems to have overlap in tumor metastasis. Snail and Slug are two transcriptional repressors of E-cadherin induced by TGFβ signaling. Snail (with functional zinc finger and SNAG domains) works to repress E-cadherin expression by creating a complex with SMAD 3/4 and binding to three E-box promoters for E-cadherin to block transcription, with the farthest downstream box having the most repressive function, leading to decreased levels of E-cadherin expression in the cell and induction of EMT [15,18]. Similarly, in vitro, Slug has also been shown to be repress E-cadherin through mediation by TGFβ through production of Sp1 transcription factor that binds to the promoter of Slug [131].
Epigenetics

Epigenetic modification plays a role in the regulation of e-cadherin. In the case of cancer, epigenetic modifications are used to repress the expression of e-cadherin in the cell, showing methylation of the CpG island in the 5′ promoter region of the e-cadherin gene [132]. In basal-like breast cancer, ZEB1 recruits DNMT1 to the promoter for CDH1, acting as an epigenetic modulator by maintaining methylation status and downregulate the presence of e-cadherin [73]. Snail acts in a similar fashion as ZEB1 by attaching to the E-box region of CDH1, but it can also working with histone methyltransferases to modulate the methylation and expression of E-cadherin [73]. Acetylation also occurs through the work of Snail by recruiting HDAC1 and HDAC2 to the CDH1 promoter and deacetylating the H3 and H4 histones [73]. Methylation and acetylation play a big role in E-cadherin expression by modulating access to the promoter sequence.
Post-Transcriptional

Post-transcriptional modification of E-cadherin can occur through the inability of E-cadherin mRNA moving to processing bodies for translation. RNA biding proteins CUG-binding protein 1 (CUGBP1) and HU antigen 1 (HUR) bind to the 3′ end of the E-cadherin mRNA, and while CUGBP1 increases E-cadherin association with processing bodies for translation to increase E-cadherin expression, HUR decreases this association and downregulates E-cadherin expression [133].
Post-Translational

E-cadherin is involved in downstream signaling in the cell through interactions between the E-cadherin cytoplasmic tail and several catenins that create a complex at adherens junctions [128]. The Src family of kinases (SFK) are also involved in the regulation of E-cadherin and catenin through phosphorylation, either phosphorylating e-cadherin which promotes its degradation or phosphorylating β-catenin which interrupts e-cadherin and α-catenin interactions, leading to a down-regulation of adhesion between adjacent cells [124]. Tyrosine phosphorylation by Src kinase of e-cadherin allows an interaction with Hakai, an E3 ubiquitin ligase that binds e-cadherin, ubiquitinating the e-cadherin which induces endocytosis of the molecule, though the exact process from ubiquitination to endocytosis is not clear [134,135]. Regulation of β-catenin through SFKs occur through the phosphorylation of β-catenin which disrupts its association with both E-cadherin and α-catenin, so cell junctions lose communication properties with the actin environment of the cell [136]. Direct contact with the e-cadherin molecule and the lectin galectin 7 facilitates regulation of e-cadherin as galectin-7 acts as stabilizing molecule in cell movement by inhibiting endocytosis of e-cadherin, so communication with neighboring cells continues with epithelial reorganization and migration such as during repair after epithelial injury [137]. In cellular adhesion, EpCAM has been shown to be a negative regulator of E-cadherin in epithelial cells by decreasing the stability of E-cadherin adhesions through disruption of E-cadherin interactions with α-catenin, promoting a more motile phenotype [113]. E-cadherin plays a role in deactivation of Hippo signaling for cell proliferation through cell–cell contact and sets off a phosphorylation event of the YAP protein in the Hippo pathway that represses proliferation, meaning destabilization of E-cadherin through EpCAM interactions could lead to activation of Hippo signaling and increased cell proliferation such as in the cancer cell phenotype [138].

### 4.2. Laminin

In addition to EpCAM and E-cadherin, laminin-1 is observed on the surface of epithelial cells, especially during embryonic development, serving as a biomarker of epithelial cell state [139]. Laminin-1 is composed of α1, β1, and γ1 chains and is typically located in the basement membrane of epithelial cells [140,141]. Laminin is regulated by the Rho/Rho kinase pathway, showing laminin-1 presence on epithelial cells and non-motile epithelial phenotypes on cells where the Rho/Rho Kinase pathway is activated [141].

Not much is known about the regulation of Laminins, showing a gap in knowledge in this area of research in the control and influences of EMT. Transcriptionally, Laminin β-1 is regulated through the ERK pathway through c-Jun because c-Jun is regulated by the ERK pathway and c-Jun binds to the promoter region of LAMB1, acting as a transcription factor to promote the expression of Laminin beta 1 [142]. Some laminins are also regulated by the NF-*κ*B signaling pathway because after activation of the pathway by TNF-α, LAMA5 transcription increased, while the transcription of LAMA4 was unaffected, but the laminin subunits of interest, alpha 1 and beta 1, are not affected in the same fashion [143]. Post-translationally, β-1, galactosyltransferase (β4GalT1) has been shown to be a regulator of β1 integrin, an important part of the structure of laminin, subjecting laminin 1 to the regulation by β4GalT1 through glycosylation [144].

Integrin β1 is also involved in in control of the ECM and epithelial differentiation as shown through the effect of the absence of integrin β1 impairing STAT5 signaling, causing epithelial cells to detach from the basement membrane [145]. In addition, laminin is also involved in organogenesis and early development processes, likely due to their interactions on the cell surface [145].

## 5. RKIP: A Novel Oncogenic EMT Suppressor

We and others have recently identified RAF-1 kinase inhibitor protein (RKIP), as a critical suppressor of metastasis and oncogenic EMT in various tumor models through interactions with known EMT regulators. as discussed below. RKIP, also known as phosphatidylethanolamine binding protein (PEBP), is an inhibitor protein that acts on the RAF signaling pathway, which is essential to EMT in tumor progression [146]. In particular, RKIP is known to play a role in the MAPK and NF-*κ*B signaling pathways, but loss of RKIP can lead to the progression of cancer and other diseases [147]. RKIP was originally found in bovine brain and is a small cytosolic protein that is highly conserved across species such as monkeys, chickens, rats, and humans [148,149]. The human homologue of RKIP is 1434 nucleotides long and shares a considerable overlap with the bovine and rat homologues, alluding to the conserved nature of RKIP across species [150].

As a major suppressor of several cancer and metastasis-promoting signaling pathways, RKIP has been considered a putative prognostic indicator and therapeutic target in cancer research, as described below in detail. Supportive to this notion are findings showing that RKIP expression is typically downregulated in many cancer types, thus allowing activation of critical pathways involved in cancer development and progression [149].

### 5.1. RKIP-Mediated Signaling in Cancer

RKIP mediates its suppressive action in the constitutive activation of several cancer progression-associated signaling pathways including the Raf-1/MEK/ERK, G-protein coupled receptor (GPCR) and NF-*κ*B pathways. Activation of the Raf-1/MEK/ERK cascade is essential for cell growth and proliferation through the epidermal growth factor receptor stimulation and corresponding Ras activation that can interact with RAF-1 [151,152]. RAF kinase phosphorylates MEK in response to signaling, continuing the cascade toward activating ERK which can relocate to the nucleus and affect gene expression [151,152]. RKIP plays a role in cell proliferation signaling through its interaction with RAF by disrupting its ability to phosphorylate and activate MEK in the RAF-1/MEK/ERK pathway [153,154]. In the regulation of the RAF-1/MEK/ERK pathway, RKIP plays a duel role in disrupting the interaction between RAF and MEK by competitively binding and inhibiting MEK phosphorylation and by binding to the N-region of RAF-1, thus inhibiting its initial activation [149]. RKIP can disrupt the signaling pathway from activating ERK by physically interacting with MEK and RAF in the pathway. PKC can phosphorylate RKIP at serine 153, and the phosphorylated RKIP dissociates from its interaction with RAF1, removing the inhibition of the MEK-ERK pathway [155]. Amplification of the signal occurs at the RAF-1/MEK interaction stage, meaning that this step is key in regulating the cascade, making RKIP an essential piece to inhibiting the signaling pathway through its competitive interaction [156,157].

The phosphorylation of RKIP disrupts RKIP binding RAF-1/MEK and instead promotes G-protein-coupled receptor (GPCR) signaling via a mechanism first described by Lorenz et al. GPCR signaling is increased at the serine 153 phosphorylated RKIP interacts with GRK-2, inhibiting its function which normally mediates GPCR phosphorylation [158]. Phosphorylation by GRK2 stops GPCR signaling. GPCRs respond to outside signaling as membrane receptors, and activated GPCRs are inhibited by arrestins, which are recruited by GRK2 after it phosphorylates the receptor [159]. The phosphorylation of RKIP is mediated by proline 74 in the ligand binding pocket, increasing the ERK signaling as phosphorylation is encouraged at serine 153 [160]. The phosphorylated RKIP interacts with GRK2 instead of RAF1, disrupting the function of GRK2, so it cannot inhibit the GPCR signaling pathway, allowing the signaling to continue and blocking the internalization of the receptor [158,161]. The switch in RKIP target with phosphorylation by PKC also changes the function of RKIP from a tumor suppressor moderating the MEK-ERK pathway to a tumor supporter function [160].

RKIP further mediates the negative regulation of NF-*κ*B signaling in cancer. RKIP acts upstream of the regulatory kinase complex that controls the inhibitor of NF-*κ*B, and directly interacts with the kinases involved in the activation of the NF-*κ*B pathway [162]. RKIP did not interact with all kinases in the activation of NF-*κ*B, specifically inhibiting NIK and TAK1 in the pathway, instead of a promiscuous interaction with all kinases to inhibit NF-*κ*B activation and inhibit the inflammatory response [162]. RKIP acts independently of the other known NF-*κ*B inhibitors, namely I*κ*B and A20, showing a complicated role of NF-*κ*B inhibition that must be more thoroughly investigated in order to determine the complexity of interactions that regulate NF-*κ*B signaling [162].

### 5.2. RKIP Expression in Human Cancers

RKIP is involved in several pathways that regulate the progression of cancer, acting as a tumor suppressor, and as a result, RKIP should be looked at as a possible target for cancer treatment. RKIP is commonly downregulated in cancers, allowing it to be used as a biomarker for metastasis as it is negatively correlated with metastatic risk in cancer progression [163]. In instances of tumors where RKIP is overexpressed, such as in multiple myeloma, the mutated form of serine 153 phosphorylated RKIP is the prominent form present, unable to inactivate the ERK pathway since it lacks the ability to interact with RAF1 [164]. In cancer, RAF1/MEK/ERK signaling is often lacking normal regulation through RKIP, since this signaling pathway controls mechanisms of cell proliferation, differentiation, and migration [165]. As discussed previously, the serine 153 phosphorylated RKIP loses its ability to interact with RAF1 and regulate the ERK pathway, allowing cancer cells to utilize the cells’ own mechanisms of proliferation and migration, and similarly, a lack of RKIP also allows the ERK signaling pathway to remain active and deregulated.

As RKIP could serve to inhibit tumor supporting pathways in cancer, some treatments aim to increase RKIP expression in tumor cells. One possible way aims to increase the RKIP mRNA, allowing for increased expression of its protein after translation. This can be mediated using the drugs 5-AzaC and DETANONO, but the lack of specificity of these drugs can have unknown biological impacts on the cell, other than increasing mRNA levels, making them an interesting place for further investigation for the effect of RKIP though their other effects may make them unusable as a cancer treatment [166]. In addition, as discussed in previous sections, regulation of RKIP expression occurs at several levels, so even though we may be able to induce higher levels of mRNA, this may not result in an increased expression of the protein, if the tumor utilizes post-transcriptional and post-translational methods of regulation. Further research could investigate how one can control the regulation of RKIP if RKIP mRNA appear to be a promising method for cancer treatment in the future.

### 5.3. Regulation of RKIP Expression

Epigenetic Regulation

Hypermethylation of the RKIP promoter sequence can regulate its protein expression [167]. In various types of advanced stage cancers, higher levels of RKIP promoter methylation correlated to a lower level of RKIP expression, and when comparing esophageal and gastric cancer methylation at this site to the normal tissue, cancerous tissues had an increased level of methylation. These finding suggest that RKIP methylation is of particular interest in the prognosis and treatment of tumors [168]. Histone modifications also play an epigenetic regulatory role in RKIP expression with histone deacetylases affecting its levels. In particular, histone deacetylase inhibitors have been shown to increase RKIP mRNA levels in some cancer types; however the opposite affect has also been observed, likely due to the activation of BTB domain and CNC homolog 1 (BACH1) by the histone deacetylase, which acts as a transcription factor that suppresses the expression of RKIP [168]. In addition to histone modifications, methylation plays a complicated role in the regulation of RKIP, and further research is needed to determine how these two processes interact on the epigenetic level to control RKIP, as well as the complicated relationship of histone deacetylases.
Transcriptional Regulation

The transcriptional regulation of RKIP takes place through several pathways. In bladder cancer, this has been noticed as a decrease in the RKIP mRNA, resulting in the downregulation of functional RKIP [169]. Androgen acts as an endocrine regulator of RKIP through androgen receptor modification of the promoter for RKIP, blocking its transcription [170]. While RKIP has been shown to have a repressive regulatory effect on Snail, conversely, Snail has been shown to have a regulatory function on RKIP as well, decreasing RKIP transcription via targeting the E-box sites that are found near the RKIP promoter, eventually blocking RKIP transcription and regulating its expression [160,171].
Post-Transcriptional Regulation

miRNAs play a significant role in RKIP regulation post-transcriptionally. miRNA-27a is involved in the knockdown of RKIP which contributes to a mesenchymal phenotype with the corresponding upregulation of vimentin and downregulation of E-cadherin, both being important in cancer progression and chemotherapy resistance [172]. In prostate cancer, miR-543 also acts as a direct post-transcriptional regulator of RKIP, via targeting and knocking down its expression and eventually, promoting cancer progression and metastasis [173]. The miRNA-mediated regulation of RKIP can have downstream effects, as in the case of miR-27a and miR-155. B-cell specific Moloney murine leukemia virus integration site 1 (Bmi-1) has been seen to have a negative relationship with RKIP expression and is associated with tumor size and prognosis of gastric cancers, but Bmi-1 does not directly influence RKIP expression. Instead, Bmi-1 induces the expression of miR-27a and miR-155 which then directly interact with RKIP to repress its expression [174].
Post-Translational Regulation

Regulation and selection of pathway for inhibition (binding of RAF-1 or GRK2) is partly based on the functional state of RKIP, switching based on the phosphorylation of S153 by Protein Kinase C (PKC), but the pocket loop of RKIP also seems to play a role in allosteric regulation of the function of RKIP [161]. The phosphorylation of RKIP by PKC at serine 153 dissociates RKIP from RAF1, allowing activation of the MEK-ERK pathway, while RKIP that is mutated to block phosphorylation at this point remains bound to RAF1, showing PKC as a functional regulator of RKIP [155,175]. The phosphorylated S153 RKIP binds to GRK2, preventing the phosphorylation of GPCR which enhances the signaling in the GPCR signaling pathway, and this signaling facilitated by RKIP has been shown to be relevant in the regulation of cardiac pressure, shown through pressure overloads in RKIP knockout mice [160]. A functional switch using a three state model has been shown to describe the RKIP switch from repression of RAF-1 to the repression of GRK2 involving both phosphorylation and pocket loop binding, although the exact role of the pocket loop in this switch needs additional study [161].

## 6. Crosstalk between RKIP and Known EMT Regulators

### 6.1. RKIP/Snail

RKIP and Snail have a direct relationship with one another, and therefore, the interactions between the two directly affect their function. In experiments in prostate cancer, Snail presence was increased as the cancer progressed to metastatic, an essential step in EMT as Snail directly targets E-cadherin, decreasing its prominence [171]. At the same time, RKIP expression also decreases, showing a relationship between the two proteins attributed to Snail binding to the E-box region of the RKIP promoter to directly repress RKIP expression transcriptionally [171]. As Snail is upregulated in EMT, the repression of RKIP should also occur with EMT in cancer progression. The MEK-ERK pathway is a target of RKIP, which acts as an inhibitor between RAF and MEK, disrupting the signaling pathway. In addition, Snail is a downstream effector of the MEK-ERK pathway, providing another connection between RKIP and Snail [176]. Inhibition of ERK signaling resulted in diminished Snail expression [176], revealing a possible mutual repression relationship between Snail and RKIP as Snail has been shown to downregulate the expression of RKIP through transcriptional regulation, but RKIP acts as an inhibitor of Snail through the inhibition of one of the Snail activation pathways. As Snail is involved with the migratory properties of cancer through EMT, RKIP is considered to be a tumor suppressor working in opposition to Snail and cancer metastasis. Snail is essential to EMT as described throughout this text, especially important in the down regulation of E-cadherin, making the mutual inhibition properties of RKIP and Snail a point of interest in cancer treatment, possibly using RKIP overexpression to limit migration and EMT in cancer. The repressed expression of active RKIP in cancer lineages shows an antagonistic relationship between RKIP and the products of EMT such as Snail.

Both RKIP and Snail are involved in a known regulation loop, the NF-*κ*B/Snail/YY1/RKIP/PTEN loop, that when dysregulated, contributes to the development of cancer phenotypes and has been linked to autophagy in cells [177]. The upstream inhibition of NF-*κ*B in the loop results in the inhibition of Snail and the upregulation of RKIP, connecting the two through the NF-*κ*B signaling pathway as downstream effects [177]. This loop is also involved in the resistance of cancer cells to apoptosis induced through cytotoxic drugs. In prostate cancer, sensitively to apoptosis decreased in response to RKIP downregulation and increased in response to RKIP upregulation [160]. Similarly, downregulation of Snail was followed by upregulation of RKIP and the corresponding sensitization to apoptotic drugs, pointing to a crossover between the regulation of RKIP and Snail by each other as the disappearance of Snail correlated to an increased in RKIP [160]. In the regulation loop, Snail recruits EZH2 to the promoter of RKIP, inhibiting it, but with the dysregulation of the loop at NF-*κ*B in cancer cells, Snail fails to repress RKIP as Snail is inhibited by NF-*κ*B inhibition [178]. As a result, RKIP is expressed and further promotes the inhibition of NF-*κ*B and by extension Snail, resulting in further apoptosis through cytotoxic drugs [178]. Cancer thrives in environments resistant to apoptosis, showing the crosstalk between RKIP and Snail signaling plays a role in the survival of cancer cells in response to apoptotic drugs (Figure 1).

### 6.2. RKIP/Vimentin

Vimentin and RKIP could have overlapping interactions in their roles in the promotion and repression EMT, respectively. Looking at the expression of RKIP and the corresponding response of EMT markers, RKIP can have a downstream effect on Vimentin expression, showing overlap between the roles of the two [179]. In nasopharyngeal carcinoma cells, RKIP overexpression corresponded to the downregulation of Vimentin and other EMT biomarkers, while in RKIP knockdown experiments, vimentin is upregulated, showing a relationship between the expression of vimentin in response to RKIP [179]. The repression of RKIP created an EMT-like response in the cells, producing EMT biomarkers that are indicative of metastasis potential and show the regulatory role RKIP has over EMT-induced biomarkers such as vimentin [179]. Similarly, in experiments involving lung cancer cells have shown that the knockdown of RKIP produced the production of EMT proteins such as vimentin, once again showing that vimentin exists downstream of the effect of RKIP [180]. In particular the activation of Notch1 under hypoxia, likely a result of RKIP repression, promoted the expression of vimentin and other EMT proteins, showing the inverse relationship of RKIP and vimentin through Notch1 regulation [180]. In cells overexpressed with RKIP, RKIP interacts with Notch1, preventing the proteolytic cleavage of the Notch Intracellular domain (NICD), inhibiting its ability to migrate to the nucleus and induce the production of EMT proteins such as vimentin [181]. While vimentin does not seem to have a relevant effect on the expression of RKIP, vimentin is regulated by pathways controlled by RKIP expression, revealing the downstream interactions between RKIP and vimentin expression (Figure 2).

### 6.3. RKIP/N-Cadherin

N-cadherin, as a downstream target of Snail and several signaling pathways, may cross-talk with RKIP in the regulation of EMT and tumor growth. N-cadherin is upregulated in EMT as discussed throughout this paper, and it is involved in the activation of the ERK pathway, which is potently inhibited by RKIP. N-cadherin is induced by Snail expression, which in turn translocates to the cellular membrane to promote cell cycle progression, via interaction with growth factors to induce MAPK activation [182]. In contrast, RKIP acts as a repressor of MAPK signaling, which in absence of RKIP activates the expression of its downstream target Snail, thus creating a loop of increasing N-cadherin expression. Therefore, the functions of RKIP and N-cadherin on cancer are shown to be antagonistic toward one another, as it relates to tumor growth and aggressiveness, through regulation of cancer cell cycle progression and EMT, respectively.

Similar to the mechanism described for vimentin, N-cadherin regulation is controlled through RKIP. Repression of RKIP induces the production of N-cadherin [180]. As discussed with vimentin, RKIP binds to the full length of Notch1, restricting the ability to cleave the NICD, and without the cleavage, the NICD cannot travel to the nucleus to induce the expression of EMT proteins such as N-cadherin [181]. The regulation of N-cadherin is partially controlled by the expression of RKIP, meaning the level of RKIP present has a downstream effect on EMT protein expression. N-cadherin, through the ERK pathway as well as through transcriptional regulation mechanisms, has a considerable amount of overlap between the activity of RKIP. Another signaling mechanism of regulation may be a point of crosstalk between N-cadherin and RKIP. In NSCLC, GATA6 antisense RNA1 (GATA6-AS1), a type of RNA, serves to regulate both as the overexpression of GATA-AS1 decreases the expression of N-cadherin and EMT while increasing the expression of RKIP [183]. Additional crossover may occur with STAT3 signaling which is blocked by RKIP, and although more research needs to be done to look at this pathway, it seems that GATA-AS1 could also inhibit STAT3 signaling, meaning RKIP and GATA3-AS1 play a similar role in blocking cancer proliferation through STAT3 signaling and N-cadherin [183] (Figure 3).

### 6.4. RKIP/E-Cadherin

Both RKIP and E-cadherin are tumor suppressors, and as such, there has been shown a positive correlation between the two, with both having decreased expression in cancer as invasion and metastasis increased [184]. RKIP in addition to expression correlations, also has a regulatory effect on E-cadherin, acting as an upstream activator [185]. In particular, in RKIP knockout studies, RKIP has been shown to activate RhoA, a negative regulator of breast cancer, which in turn, stabilizes E-cadherin in adherens junctions and regulates E-cadherin localization to the membrane [185]. RKIP regulates E-cadherin through Erk2 in the MEK/ERK pathway as well since RKIP inhibits the activation and phosphorylation of Erk2 [185]. Erk2 is a regulator of RhoA (and by downstream association E-cadherin) because Erk 2 disrupts the RhoA activator GEF-H1 [185]. E-cadherin could also be associated with RKIP through the other pathways that RKIP regulates such as NF-kB, which is activated in response to RKIP loss, which has an inverse relationship with E-cadherin, but more research would need to focus on the mechanism of such interaction if one exists [186].

As discussed previously in the regulation of E-cadherin, Snail can act as a repressor of E-cadherin by attaching to the E-box region of CDH1. It has also been established that RKIP acts as a repressor of Snail through inhibition of the MEK/ERK pathway, while Snail transcriptionally represses RKIP expression, showing a repressive loop between the two. Therefore, Snail could provide a point of crosstalk between RKIP and E-cadherin. As RKIP is a repressor of Snail and Snail is a repressor of E-cadherin, RKIP has downstream effects on the expression of E-cadherin. RKIP inhibits the inhibitor of E-cadherin, promoting the expression E-cadherin through its downstream effects of repressing Snail (Figure 4).

### 6.5. RKIP/EPCAM

The crosstalk between RKIP and EPCAM pathways is one area of research that is still relatively unknown. While RKIP is very involved with the MEK/ERK pathway and the NF-kB pathway, EPCAM is more involved with the construction of adherens junctions and adhesion between cells. One possible point of crosstalk between the two could be through E-cadherin. EPCAM is responsible for regulating molecules of adherens junctions such as nectin 1, and E-cadherin is a major component of adherens junctions [116]. In knockout studies of EPCAM, nectin 1 mRNA levels were greatly reduced, and nectin 1 can regulate the ability of E-cadherin to function as an adhesion molecule, possibly downregulating the functional capabilities of E-cadherin and adherens junctions [116]. With this connection between the regulation of E-cadherin and EPCAM, a cross regulation can be connected between RKIP and EPCAM. Since, as previously established in this paper, RKIP can promote the expression of E-cadherin by inhibiting the repressive function of Erk2. This shows that RKIP and EPCAM have crossover between their downstream regulation on E-cadherin, since both are involved in creating stable adherens junctions through the promotion of E-cadherin and other molecules such as nectin 1 involved in the creation of adherens junctions. Other than E-cadherin, RKIP and EPCAM have very little known crossover between their regulation or their regulatory functions, leaving a gap in knowledge about their possible crosstalk in their involvement in EMT (Figure 5).

### 6.6. RKIP/Laminin

Though the research on the crosstalk between RKIP and laminin is relatively scarce, there could be overlap through the MEK/ERK pathway. As discussed previously, RKIP inhibits the activation of the ERK pathway through its interaction with Raf. Raf is then blocked from phosphorylating MEK and continuing the signaling cascade. Therefore, RKIP is a key player in the regulation of the MEK/ERK pathway. Laminin has been found, as discussed previously, to be regulated by the ERK pathway because the ERK pathway controls the regulation of c-Jun. c-Jun binds to the promoter region of LAMB1, promoting the expression of laminin-1. Therefore, RKIP could play a key role in the expression of laminin through its ability to regulate the MEK/ERK pathway and inhibit the signaling cascade that regulates c-Jun. Though this crosstalk between the RKIP and laminin seems to be present, there is little research on how these two proteins actually interact with one another, and this crosstalk through the MEK/ERK pathway is only one possibility that could occur through RKIP downstream regulating laminin presence (Figure 6).

## 7. Association Patterns between RKIP Expression and Known EMT Regulators Assessed by Bioinformatic Analyses

We extracted RNA-seq data (read counts) for the genes PEBP1 (RKIP), SNAI1 (SNAIL), VIM (Vimentin), CDH1 (E-Cadherin), CDH2 (N-Cadherin), EPCAM, LAMA1 and LAMB1 (laminin subunits alpha 1 and beta 1, respectively), from the Cancer Genome Atlas (TCGA) using the Genomic Data Commons (GDC) Data Portal (https://portal.gdc.cancer.gov/) (accessed on 8 July 2022). Read counts were then normalized to log_2_ transcripts per million (TPM) mapped reads adding an offset of 1, as previously described [187]. Gene expression-related comparisons were made between tumor and normal samples, across 22 different types of cancer in the TCGA database and a *p*-value <0.05 was considered threshold for statistical significance. The results showed a great diversity in the expression of all PEBP1, as well as the rest genes of interest, as previously reported [154]. In addition, we analyzed the correlation pair-wise (Pearson’s correlation test), between PEBP1 (RKIP) and SNAI1, VIM, CDH1/2, EPCAM, and LAMA1/B1 genes, across the 22 different types of cancer and normal tissues in the TCGA database. The log_2_ TPM scale of gene expression was used for visualization of the pair-wise correlations. Despite the huge diversity across all different types of cancer, we noticed strong correlations between the expression levels of PEBP1 and SNAI1, PEBP1 and VIM, as well as between PEBP1 and EPCAM 

Examining the EMT gene products it was determined that there was an inverse relationship between the expression of RKIP and Snail and vimentin, and while there was a positive correlation with E-cadherin, N-cadherin, and EPCAM. Laminin α subunit shows either a direct or indirect relationship with RKIP depending on cancer type, while laminin subunit β shows dominantly an inverse relationship. The signaling cross-talks between RKIP and these various gene products are well presented in detail in the text and summarized in schematic diagrams. Briefly below, we present the highlights of these:
(1)For RKIP and Snail, for example, while Snail is a transcriptional repressor of RKIP, in turn, RKIP inhibits NF-kB and downstream its target Snail. IN addition, the inhibition of the Raf/Mek/Erk pathway by RKIP results in the inhibition of downstream effector Snail. (Figure 1) Bioinformatic analyses using the pair-wise Pearsons correlations across 23 different types of human cancers and normal tissues revealed that RKIP was negatively correlated with Snail in 9 cancer types and positively correlated in 2 cancer types (Table 1, Appendix A).(2)For vimentin, there was an inverse relationship between the expressions of RKIP and vimentin experimentally in various cancers. For example, the activation of Notch 1 which promotes the expression of vimentin and RKIP’s interaction with Notch 1 prevents its nuclear localization and the expression of vimentin amongst other EMT proteins (Figure 2). Bioinformatic analyses showed that RKIP was negatively correlated with vimentin in 10 cancer types (as expected) and positively correlated with 4 cancer types (Table 2, Appendix A).(3)For N-cadherin, there was an inverse relationship between RKIP and N-cadherin expressions. N-cadherin is a target of upstream Snail and Snail is inhibited by RKIP, hence inhibition f N-cadherin by RKIP. Additionally, vimentin activates the Erk pathway which regulates Snail and thus the inhibition by RKIP of the ERK/Snail/N-cadherin axis. (Figure 3). Bioinformatic analyses demonstrated that RKIP was negatively correlated with only one cancer type and positively correlated with 7 cancer types (Table 3, Appendix A). Interestingly, these data are not predicted nor expected and reveal that each cancer type signaling network is different and complex and the various cross-talks are being modulated by different factors inherent with the cancer type.(4)The overexpression of RKIP results in the inhibition of NF-kB and downstream the RKIP repressor Snail Figure 4). With E-cadherin, there was a positive correlation with 14 cancer types (as expected) and negative correlation with one cancer type. (Table 4, Appendix A).(5)RKIP and EPCAM play a role in the stabilization of E-cadherin at the adherens junctions (Figure 5). With EPCAM, there were positive correlations with 14 cancer types (as expected) and negatively correlated with one cancer type (Table 5, Appendix A).(6)RKIP regulates laminin alpha 1 via the activation of c-jun (Figure 6). For laminin subunit alpha 1, there were 6 positive correlations with 6 cancer types and inverse correlations with 5 cancer types (Table 6, Appendix A).(7)With laminin subunit beta 1, there was negative correlations with 7 cancer types and positive correlations with 5 cancer types (Table 7, Appendix A).

## 8. Concluding Remarks

Since the original discovery of RKIP in 1999, and its unique molecule to interact and inhibit the Raf/Mek/Erk signaling and protumorigenic pathway, significant new findings were reported of the pleotropic activities RKIP play in normal and several diseases particularly in cancers. RKIP expression is significantly reduced in many cancers compared to adjacent normal tissues and more reduction in RKIP levels in metastases. One of the original findings was the demonstration that RKIP is involved in the metastatic process and, experimentally in vivo in mice, its overexpression in cancer lines inhibited the tumor metastatic potential. Therefore, RKIP was called a “metastasis suppressor”. Our published reports and others further demonstrated that RKIP overexpression inhibited the EMT phenotype, namely, inhibiting the expression of EMT mesenchymal gene products such as N-cadherin, Snail. vimentin, and EPCAM while upregulated the expressions of the epithelial phenotype gene products such as E-cadherin and Laminin. In contrast, RKIP inhibition potentiated the EMT phenotype. These findings suggested that RKIP is involved in the positive regulation of these epithelial and negatively the mesenchymal gene products. It was known of the underlying mechanisms by which RKIP regulates these gene products and we hypothesized and inferred from these findings that RKIP regulation may be the results of signaling cross-talks between RKIP and both the epithelial and mesenchymal gene products mentioned above. This review analyzed and verified the existence of these various signaling cross-talks which were also validated, in large part, by bioinformatic analyses of various human cancer tissues from available data sets. 

From the bioinformatic analyses, clearly, the expected inverse relationship between RKIP and the mesenchymal EMT gene products as well as the direct relationship of RKIP with the epithelial gene products were generally consistent in many of the cancers analyzed. However, the analyses also revealed that for certain gene products there were no observed relationships one way or another as the findings were not significant. These findings explain the low numbers of cancers with a relationship amongst the total number examined. In addition, there were findings in some cancers that did not follow the expected results. It is not clear of the underlying mechanisms responsible for these findings that may have to do with the complex signaling networks and regulation (transcriptional, epigenetics, post-transcriptional and translational) that are unique within each tumor type. Further studies are needed to unravel these mechanisms.

Overall, our findings support the pivotal role mediated by RKIP, as a metastasis suppressor and inhibitor of EMT, in the regulation of various gene products involved in the progression of cancer cells from the epithelial phenotype to the mesenchymal phenotype. Hence, the low expression of RKIP in many cancers promotes the expression of several gene products that are responsible for EMT and metastases. Thus, it is attractive to develop means for induction of the expression of RKIP in cancers in an effort to inhibit the EMT program and metastasis. In addition, the induction of RKIP expression will also lead to sensitize the resistant tumor cells to respond to both chemotherapy and immunotherapy-mediated cell death resulting in tumor regression along with a significant prolongation of survival.

## Figures and Tables

**Figure 1 cancers-14-04596-f001:**
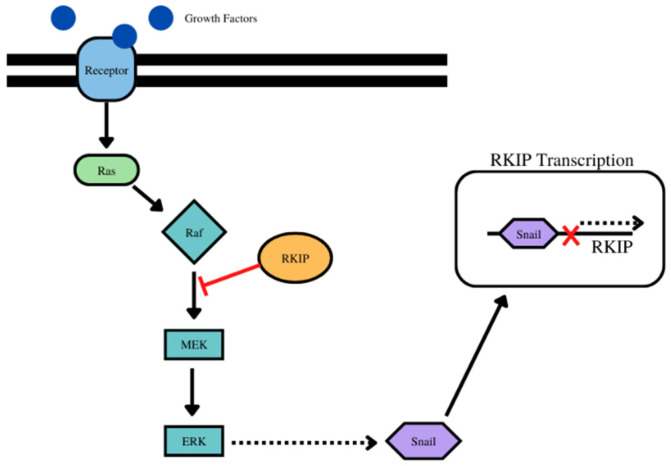
Crosstalk Between RKIP and Snail. RKIP inhibits the MEK/ERK signaling pathway by interrupting the interaction between Raf and MEK, stopping the progression of the signal through the pathway. Snail is regulated downstream of the MEK/ERK pathway, upregulated through MEK/ERK signaling. In addition, Snail is an inhibitor of RKIP transcription through binding to the promoter region of RKIP.

**Figure 2 cancers-14-04596-f002:**
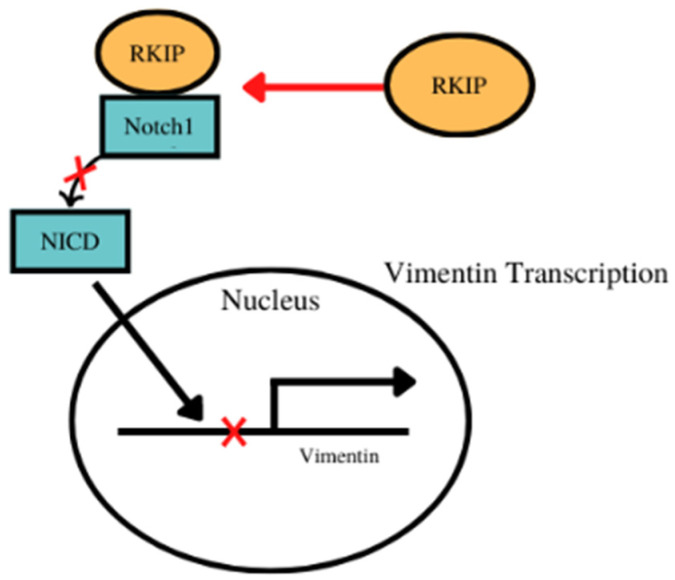
Crosstalk Between RKIP and Vimentin. RKIP binds to Notch1, blocking the cleavage of NICD. NICD is then unable to locate to the nucleus to induce the transcription of vimentin. RKIP is able to regulate the transcription of vimentin through interaction with Notch1.

**Figure 3 cancers-14-04596-f003:**
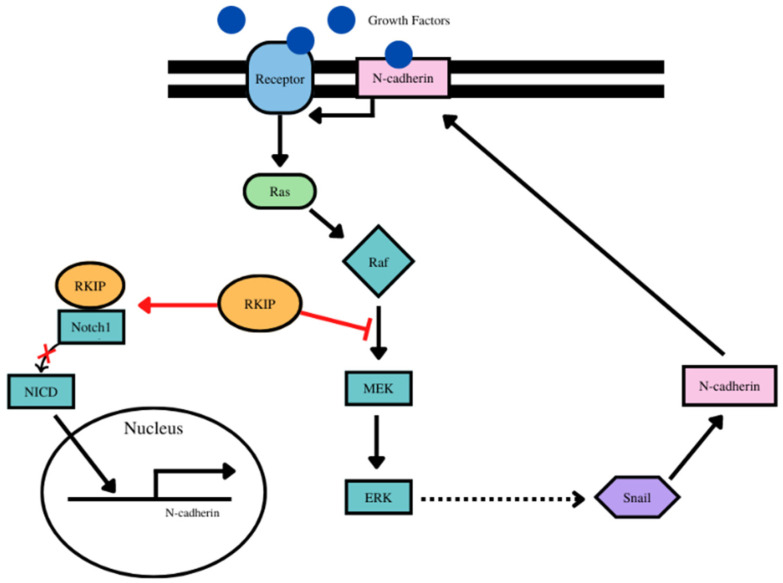
Crosstalk Between RKIP and N-cadherin. RKIP inhibits the MEK/ERK signaling pathway by interrupting the interaction between Raf and MEK, stopping the progression of the signal through the pathway. Snail is regulated downstream of the MEK/ERK pathway, upregulated through MEK/ERK signaling. Snail cannot promote N-cadherin production without MEK/ERK signaling, showing RKIP as an antagonist to N-cadherin expression. In addition, RKIP binds to Notch1, blocking the cleavage of NICD. NICD is then unable to locate to the nucleus to induce the transcription of N-cadherin.

**Figure 4 cancers-14-04596-f004:**
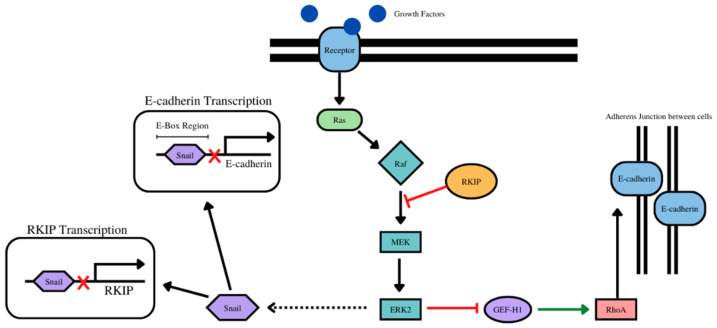
Crosstalk between RKIP and E-cadherin. RKIP inhibits the MEK/ERK signaling pathway by interrupting the interaction between Raf and MEK, stopping the progression of the signal through the pathway. Erk 2 is an inhibitor of GEF-H1, an activator of RhoA which helps in the stabilization of E-cadherin at adherens junctions. ERK signaling increases Snail production which inhibits the transcription of both RKIP and E-cadherin.

**Figure 5 cancers-14-04596-f005:**
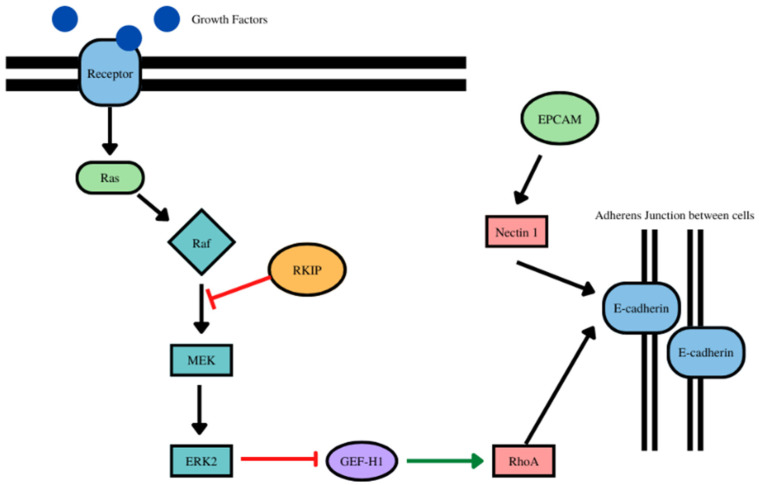
Crosstalk Between RKIP and EPCAM. RKIP inhibits the MEK/ERK signaling pathway by interrupting the interaction between Raf and MEK, stopping the progression of the signal through the pathway. Erk 2 is an inhibitor of GEF-H1, an activator of RhoA which helps in the stabilization of E-cadherin at adherens junctions. EPCAM regulates nectin1 which plays a role in E-cadherins ability to function as an adhesion molecule.

**Figure 6 cancers-14-04596-f006:**
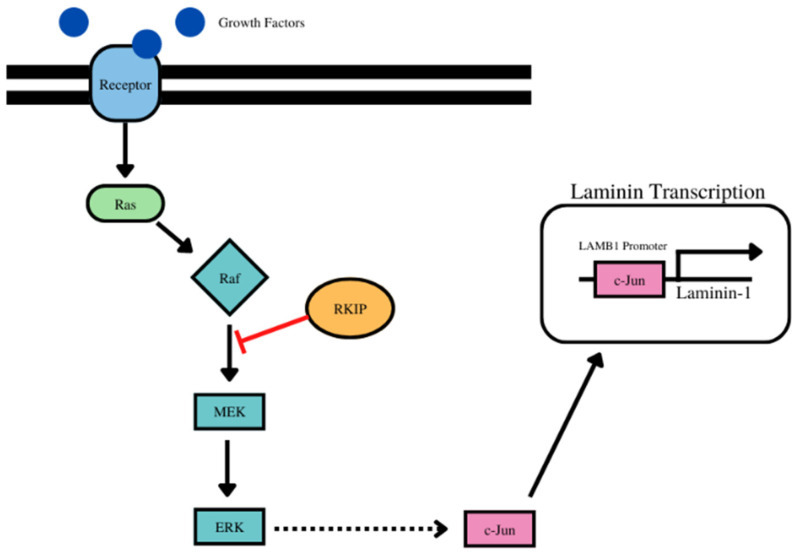
Crosstalk Between RKIP and Laminin. RKIP inhibits the MEK/ERK signaling pathway by interrupting the interaction between Raf and MEK, stopping the progression of the signal through the pathway. C-Jun is regulated downstream of the MEK/ERK pathway. C-Jun binds to the promoter of Laminin-1, regulating the expression of Laminin-1.

**Table 1 cancers-14-04596-t001:** RKIP vs. Snail. A bioinformatic analysis was performed across 23 different cancer types comparing the presence of RKIP against Snail. From this data, a *p*-value and an R value were determined for each of the cancer types. The table above organized the bioinformatic data. For each cancer type, the R value was reported in the third column. Significance was determined based on the *p*-value, with a *p*-value less than 0.05 indicating a significant value. The relationship between RKIP and Snail was determined through the sign of the R value. A negative R value indicated an inverse relationship while a positive R value indicated a direct relationship. The rows highlighted in blue indicate a significant, inverse relationship between RKIP and Snail. The rows highlighted in red indicate a significant, direct relationship between RKIP and Snail.

SNAIL vs. RKIP				
Cancer	Gene vs. RKIP	R Values	Significance	Relationship
BLCA	SNAI1	−0.14	S	Inverse
BRCA	SNAI1	−0.2	S	Inverse
CESC	SNAI1	−0.008	NS	Inverse
CHOL	SNAI1	−0.013	NS	Inverse
COAD	SNAI1	−0.017	NS	Inverse
ESCA	SNAI1	0.0056	NS	Direct
HNSC	SNAI1	−0.013	NS	Inverse
KICH	SNAI1	−0.29	S	Inverse
KIRC	SNAI1	−0.18	S	Inverse
KIRP	SNAI1	−0.19	S	Inverse
LICH	SNAI1	−0.26	S	Inverse
LUAD	SNAI1	−0.18	S	Inverse
LUSC	SNAI1	−0.055	NS	Inverse
PAAD	SNAI1	0.071	NS	Direct
PCPG	SNAI1	−0.11	NS	Inverse
PRAD	SNAI1	−0.15	S	Inverse
READ	SNAI1	0.15	NS	Direct
SARC	SNAI1	−0.038	NS	Inverse
SKCM	SNAI1	−0.15	S	Inverse
STAD	SNAI1	0.11	S	Direct
THCA	SNAI1	0.44	NS	Direct
THYM	SNAI1	0.21	S	Direct
UCEC	SNAI1	−0.085	NS	Inverse

Blue: Significant Inverse Relationship; Red: Significant Direct Relationship.

**Table 2 cancers-14-04596-t002:** RKIP vs. VIM. A bioinformatic analysis was performed across 23 different cancer types comparing the presence of RKIP against VIM. From this data, a *p*-value and an R value were determined for each of the cancer types. The table above organized the bioinformatic data. For each cancer type, the R value was reported in the third column. Significance was determined based on the *p*-value, with a *p*-value less than 0.05 indicating a significant value. The relationship between RKIP and VIM was determined through the sign of the R value. A negative R value indicated an inverse relationship while a positive R value indicated a direct relationship. The rows highlighted in blue indicate a significant, inverse relationship between RKIP and VIM. The rows highlighted in red indicate a significant, direct relationship between RKIP and VIM.

VIM vs. RKIP				
Cancer	Gene vs. RKIP	R Values	Significance	Relationship
BLCA	VIM	−0.15	S	Inverse
BRCA	VIM	−0.25	S	Inverse
CESC	VIM	0.25	S	Direct
CHOL	VIM	−0.068	NS	Inverse
COAD	VIM	−0.15	S	Inverse
ESCA	VIM	−0.2	S	Inverse
HNSC	VIM	−0.011	NS	Inverse
KICH	VIM	−0.16	NS	Inverse
KIRC	VIM	−0.13	S	Inverse
KIRP	VIM	0.037	NS	Direct
LICH	VIM	−0.29	S	Inverse
LUAD	VIM	−0.1	S	Inverse
LUSC	VIM	−0.025	NS	Inverse
PAAD	VIM	0.068	NS	Direct
PCPG	VIM	−0.16	S	Inverse
PRAD	VIM	−0.11	S	Inverse
READ	VIM	−0.12	NS	Inverse
SARC	VIM	−0.059	NS	Inverse
SKCM	VIM	0.2	S	Direct
STAD	VIM	−0.11	S	Inverse
THCA	VIM	0.21	S	Direct
THYM	VIM	0.47	S	Direct
UCEC	VIM	0.13	NS	Direct

Blue: Significant Inverse Relationship; Red: Significant Direct Relationship.

**Table 3 cancers-14-04596-t003:** RKIP vs. CDH2. A bioinformatic analysis was performed across 23 different cancer types comparing the presence of RKIP against CDH2. From this data, a *p*-value and an R value were determined for each of the cancer types. The table above organized the bioinformatic data. For each cancer type, the R value was reported in the third column. Significance was determined based on the *p*-value, with a *p*-value less than 0.05 indicating a significant value. The relationship between RKIP and CDH2 was determined through the sign of the R value. A negative R value indicated an inverse relationship while a positive R value indicated a direct relationship. The rows highlighted in blue indicate a significant, inverse relationship between RKIP and CDH2. The rows highlighted in red indicate a significant, direct relationship between RKIP and CDH2.

CDH2 vs. RKIP				
Cancer	Gene vs. RKIP	R Values	Significance	Relationship
BLCA	CDH2	−0.067	NS	Inverse
BRCA	CDH2	−0.044	NS	Inverse
CESC	CDH2	0.11	NS	Direct
CHOL	CDH2	0.24	NS	Direct
COAD	CDH2	−0.11	NS	Inverse
ESCA	CDH2	0.064	NS	Direct
HNSC	CDH2	−0.015	NS	Inverse
KICH	CDH2	−0.17	NS	Inverse
KIRC	CDH2	0.32	S	Direct
KIRP	CDH2	0.26	S	Direct
LICH	CDH2	0.039	NS	Direct
LUAD	CDH2	−0.017	NS	Inverse
LUSC	CDH2	0.16	S	Direct
PAAD	CDH2	0.33	S	Direct
PCPG	CDH2	0.17	S	Direct
PRAD	CDH2	−0.0097	NS	Inverse
READ	CDH2	0.3	S	Direct
SARC	CDH2	0.12	NS	Direct
SKCM	CDH2	−0.13	S	Inverse
STAD	CDH2	0.059	NS	Direct
THCA	CDH2	0.34	S	Direct
THYM	CDH2	0.058	NS	Direct
UCEC	CDH2	−0.01	NS	Inverse

Blue: Significant Inverse Relationship; Red: Significant Direct Relationship.

**Table 4 cancers-14-04596-t004:** RKIP vs. CDH1. A bioinformatic analysis was performed across 23 different cancer types comparing the presence of RKIP against CDH1. From this data, a *p*-value and an R value were determined for each of the cancer types. The table above organized the bioinformatic data. For each cancer type, the R value was reported in the third column. Significance was determined based on the *p*-value, with a *p*-value less than 0.05 indicating a significant value. The relationship between RKIP and CDH1 was determined through the sign of the R value. A negative R value indicated an inverse relationship while a positive R value indicated a direct relationship. The rows highlighted in blue indicate a significant, inverse relationship between RKIP and CDH1. The rows highlighted in red indicate a significant, direct relationship between RKIP and CDH1.

CDH1 vs. RKIP				
Cancer	Gene vs. RKIP	R Values	Significance	Relationship
BLCA	CDH1	0.28	S	Direct
BRCA	CDH1	0.23	S	Direct
CESC	CDH1	0.068	NS	Direct
CHOL	CDH1	0.39	S	Direct
COAD	CDH1	0.2	S	Direct
ESCA	CDH1	−0.036	NS	Inverse
HNSC	CDH1	0.19	S	Direct
KICH	CDH1	0.11	NS	Direct
KIRC	CDH1	0.12	S	Direct
KIRP	CDH1	0.018	NS	Direct
LICH	CDH1	−0.16	S	Inverse
LUAD	CDH1	0.087	NS	Direct
LUSC	CDH1	0.13	S	Direct
PAAD	CDH1	−0.053	NS	Inverse
PCPG	CDH1	−0.098	NS	Inverse
PRAD	CDH1	0.3	S	Direct
READ	CDH1	0.23	S	Direct
SARC	CDH1	0.014	NS	Direct
SKCM	CDH1	0.28	S	Direct
STAD	CDH1	0.099	S	Direct
THCA	CDH1	0.42	S	Direct
THYM	CDH1	0.7	S	Direct
UCEC	CDH1	0.33	S	Direct

Blue: Significant Inverse Relationship; Red: Significant Direct Relationship.

**Table 5 cancers-14-04596-t005:** RKIP vs. EPCAM. A bioinformatic analysis was performed across 23 different cancer types comparing the presence of RKIP against VIM. From this data, a *p*-value and an R value were determined for each of the cancer types. The table above organized the bioinformatic data. For each cancer type, the R value was reported in the third column. Significance was determined based on the *p*-value, with a *p*-value less than 0.05 indicating a significant value. The relationship between RKIP and EPCAM was determined through the sign of the R value. A negative R value indicated an inverse relationship while a positive R value indicated a direct relationship. The rows highlighted in blue indicate a significant, inverse relationship between RKIP and EPCAM. The rows highlighted in red indicate a significant, direct relationship between RKIP and EPCAM.

EPCAM vs. RKIP				
Cancer	Gene vs. RKIP	R Values	Significance	Relationship
BLCA	EPCAM	0.3	S	Direct
BRCA	EPCAM	0.098	S	Direct
CESC	EPCAM	0.25	S	Direct
CHOL	EPCAM	0.18	NS	Direct
COAD	EPCAM	0.098	NS	Direct
ESCA	EPCAM	0.26	S	Direct
HNSC	EPCAM	0.45	S	Direct
KICH	EPCAM	0.28	S	Direct
KIRC	EPCAM	0.13	S	Direct
KIRP	EPCAM	0.028	NS	Direct
LICH	EPCAM	−0.24	S	Inverse
LUAD	EPCAM	0.19	S	Direct
LUSC	EPCAM	0.23	S	Direct
PAAD	EPCAM	−0.022	NS	Inverse
PCPG	EPCAM	−0.0053	NS	Inverse
PRAD	EPCAM	0.18	S	Direct
READ	EPCAM	0.012	NS	Direct
SARC	EPCAM	0.043	NS	Direct
SKCM	EPCAM	0.0039	NS	Direct
STAD	EPCAM	0.018	S	Direct
THCA	EPCAM	0.26	S	Direct
THYM	EPCAM	0.57	S	Direct
UCEC	EPCAM	0.23	S	Direct

Blue: Significant Inverse Relationship; Red: Significant Direct Relationship.

**Table 6 cancers-14-04596-t006:** RKIP vs. LAMA1. A bioinformatic analysis was performed across 23 different cancer types comparing the presence of RKIP against VIM. From this data, a *p*-value and an R value were determined for each of the cancer types. The table above organized the bioinformatic data. For each cancer type, the R value was reported in the third column. Significance was determined based on the *p*-value, with a *p*-value less than 0.05 indicating a significant value. The relationship between RKIP and LAMA1 was determined through the sign of the R value. A negative R value indicated an inverse relationship while a positive R value indicated a direct relationship. The rows highlighted in blue indicate a significant, inverse relationship between RKIP and LAMA1. The rows highlighted in red indicate a significant, direct relationship between RKIP and LAMA1.

LAMA1 vs. RKIP				
Cancer	Gene vs. RKIP	R Values	Significance	Relationship
BLCA	LAMA1	−0.037	NS	Inverse
BRCA	LAMA1	−0.0905	S	Inverse
CESC	LAMA1	0.15	S	Direct
CHOL	LAMA1	0.0048	NS	Direct
COAD	LAMA1	−0.17	S	Inverse
ESCA	LAMA1	0.15	S	Direct
HNSC	LAMA1	0.11	S	Direct
KICH	LAMA1	0.018	NS	Direct
KIRC	LAMA1	0.27	S	Direct
KIRP	LAMA1	−0.064	NS	Inverse
LICH	LAMA1	−0.17	S	Inverse
LUAD	LAMA1	0.032	NS	Direct
LUSC	LAMA1	0.085	NS	Direct
PAAD	LAMA1	−0.15	NS	Inverse
PCPG	LAMA1	−0.16	S	Inverse
PRAD	LAMA1	0.037	NS	Direct
READ	LAMA1	−0.17	NS	Inverse
SARC	LAMA1	−0.02	NS	Inverse
SKCM	LAMA1	0.21	S	Direct
STAD	LAMA1	0.12	S	Direct
THCA	LAMA1	−0.17	S	Inverse
THYM	LAMA1	0.045	NS	Direct
UCEC	LAMA1	0.056	NS	Direct

Blue: Significant Inverse Relationship; Red: Significant Direct Relationship.

**Table 7 cancers-14-04596-t007:** RKIP vs. LAMB1. A bioinformatic analysis was performed across 23 different cancer types comparing the presence of RKIP against VIM. From this data, a *p*-value and an R value were determined for each of the cancer types. The table above organized the bioinformatic data. For each cancer type, the R value was reported in the third column. Significance was determined based on the *p*-value, with a *p*-value less than 0.05 indicating a significant value. The relationship between RKIP and LAMB1 was determined through the sign of the R value. A negative R value indicated an inverse relationship while a positive R value indicated a direct relationship. The rows highlighted in blue indicate a significant, inverse relationship between RKIP and LAMB1. The rows highlighted in red indicate a significant, direct relationship between RKIP and LAMB1.

LAMB1 vs. RKIP				
Cancer	Gene vs. RKIP	R Values	Significance	Relationship
BLCA	LAMB1	−0.09	NS	Inverse
BRCA	LAMB1	−0.12	S	Inverse
CESC	LAMB1	0.13	S	Direct
CHOL	LAMB1	−0.074	NS	Inverse
COAD	LAMB1	0.0095	NS	Direct
ESCA	LAMB1	0.088	NS	Direct
HNSC	LAMB1	0.031	NS	Direct
KICH	LAMB1	0.25	S	Direct
KIRC	LAMB1	−0.13	S	Inverse
KIRP	LAMB1	0.03	NS	Direct
LICH	LAMB1	−0.038	S	Inverse
LUAD	LAMB1	−0.14	S	Inverse
LUSC	LAMB1	0.027	NS	Direct
PAAD	LAMB1	0.22	S	Direct
PCPG	LAMB1	−0.021	S	Inverse
PRAD	LAMB1	−0.094	S	Inverse
READ	LAMB1	−0.069	NS	Inverse
SARC	LAMB1	−0.037	NS	Inverse
SKCM	LAMB1	−0.15	S	Inverse
STAD	LAMB1	−0.034	NS	Inverse
THCA	LAMB1	0.2	S	Direct
THYM	LAMB1	0.33	S	Direct
UCEC	LAMB1	−0.027	NS	Inverse

Blue: Significant Inverse Relationship; Red: Significant Direct Relationship.

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
