# Peer review of "The Role of RKIP in the Regulation of EMT in the Tumor Microenvironment"

_cancers, 2022, doi:10.3390/cancers14194596_

Round 1

Reviewer 1 Report

The review article titled “The role of RKIP in the regulation of EMT in the tumor micro-environment” by Cessana et al. talks about the RAF-kinase inhibitor protein (RKIP). RKIP directly inhibits the Raf/Mek/Erk and NF-kB pathways in cancer cells and leading to the inhibition of cell proliferation, viability, EMT, and metastasis. RKIP plays a role in regulating cancer cell resistance to chemotherapy and immunotherapy. The low expression of RKIP, inhibition of EMT, and metastasis by RKIP led to an underline mechanism of therapeutics. Therefore, developing a chemotherapeutic agent to selectively activate the expression of RKIP may inhibit tumor suppression EMT and metastasis. This review article is written comprehensively and includes almost every aspect of RKIP. The pointwise comments are as follows;

1.    What is the role of RKIP in developing extracellular matrix protein in metastasis?

2.    What are the putative interactor proteins of RKIP?

3.    What are the limitations of RKIP in cancer suppression?

Author Response

  1. The role of RKIP in developing extracellular matrix proteins in metastases? This has been reported in many publications and is not relevant at all in our review
  2. What are the putative interactor proteins if RKP? This has been addressed in our review 
  3. What are the limitations of RKIP in cancer progression? We have addressed this in our discussion 

Reviewer 2 Report

Good job! Maybe, I just would add a few more images to support such a large amount of finely tuned information.

Author Response

This reviewer was satisfied with the review and did not address any comments. There was a suggestion regarding the possibility to add more figures. However, the manuscript already has many Figures and Tables as well as supplementary figures. These represented well the contents without adding redundant figures published by others. 

Reviewer 3 Report

This review summarizes the role of RKIP in the regulation of EMT in tumor tissue. Overall, the review gives enough literature review about the function of RKIP in EMT. Some revisions are suggested.

A major one is that the context describes too much information about EMT-associated biomarkers and their regulation, which are suggested to be shorten.

Remove the space after the period of the sentences, such as line 25 and line 26.

Modify the references using the current journal type.

Check abbreviations of molecules, Cdc42.

Author Response

We thank the reviewer for the minor  comments raised. 

  1. We have shorten the requested section as recommended.
  2. We did remove the space and the period sentences as requested
  3. The references were modified as per the publisher's formatting
  4. We checked the abbreviations of cdc42